# Assessing Traffic Characteristics for Safe Pedestrian Crossings: Developing Warrants for Sustainable Urban Safety

**Shivang Chauhan [1], Sanjay Dave [2], Jiten Shah [3] and Ashu Kedia [4,*]**

1   Civil Engineering Department, NIMS Institute of Engineering and Technology, NIMS University Rajasthan, Jaipur 302131, India; shivangchauhan98@gmail.com
2   Department of Civil Engineering, The Maharaja Sayajirao University Baroda, Vadodara 390002, India; smdave-ced@msubaroda.ac.in
3   Department of Civil Engineering, Institute of Infrastructure, Technology, Research and Management, Ahmedabad 380026, India; jitenshah@iitram.ac.in
4   Urban Connection Limited, Christchurch 8011, New Zealand
*   Correspondence: ashu.kedia0209@gmail.com

**Abstract:** The escalating urbanisation fuelled by population growth and economic expansion has triggered a notable surge in vehicular and pedestrian traffic, amplifying their interaction. Nonetheless, inadequate research, investment, and prioritisation have engendered inefficient pedestrian crossing infrastructures. This study endeavours to bridge this gap by crafting tailored warrants suited to Indian traffic dynamics, facilitating the implementation of pedestrian crossing facilities. Employing $PV^2$ threshold value analysis, this study scrutinises pedestrian behavioural traits, such as gap acceptance and waiting time. Additionally, K-means clustering analysis delineates distinct levels of severity (LOSe), grounded in variables encompassing vehicular and pedestrian flow, gap acceptance, and waiting time. By establishing the nexus between vehicular volume and gap acceptance and vehicular volume and waiting time, a spectrum of $PV^2$ threshold values is delineated. These LOS categories guide the selection of pedestrian facilities, ensuring secure pedestrian–vehicle interactions. Leveraging $PV^2$ charts and vehicular volume assessments, our research identifies fitting pedestrian crossing infrastructures, thereby bolstering road safety for pedestrians and vehicles, underpinning sustainable urban mobility.

**Keywords:** urbanization; pedestrian safety; traffic dynamics; sustainable mobility; $PV^2$ analysis; urban development; level of severity (LOSe); K-means clustering; gap acceptance

## 1. Introduction

The escalating vehicular and pedestrian movements in urban areas have surged significantly due to rapid population density and economic development rise. This intensified interaction between pedestrians and vehicles has made pedestrian–vehicle conflicts the most severe issue in urban regions. Pedestrian–vehicle interaction is the most vulnerable to road crashes, which occur when a pedestrian crosses the road. Approximately 1.35 million people die due to road accidents, of which 5% are among vulnerable road users, including pedestrians, cyclists, and motorcyclists [1]. In India, there were a total of 151,113 fatalities, out of which 25,858 were pedestrians who lost their lives in road accidents [2]. With 1% of the world's vehicles, India accounts for 11% of the global deaths in road accidents [3]. These crash numbers suggest that these crashes cost around 3.14% of the country's GDP [2]. The absence of suitable crossing facilities exacerbates the severity of these incidents and stands as a leading cause of fatalities in India. According to STATS19, police data on road accidents show that 75% of pedestrian road crashes occurred where pedestrian crossing facilities were absent, while the remaining 25% occurred even when crossing facilities were provided [4]. Recognising this hazardous scenario underscores the need to provide pedestrian crossing facilities and ensure their appropriateness considering road conditions.

Mid-block crossings have been identified as the most dangerous crossing locations [5], compounded by inadequate pedestrian crossing facilities. In the USA, 17% of pedestrian fatalities occur at intersections. In comparison, 73% occur at non-intersection locations, with the remaining 10% happening at various sites, such as roadsides, shoulders, parking lanes, bicycle lanes, sidewalks, medians, crossing islands, and driveway accesses [6]. The non-compliant behaviour of motorists and the absence of lane discipline in developing countries like India contribute to the chaos at pedestrian crossings. At uncontrolled crossroads, pedestrians navigate gaps between vehicles to cross; at controlled mid-block or crossroads, pedestrians have to wait for the green signal, as more priority is given to vehicles, which leads to violating the signal and taking risks [7,8]. Pedestrians take these risks and violate signals when the wait time at the kerb exceeds 48 s [9,10]. Historically, the focus on highway transportation has prioritised enhancing the safety and mobility of motor vehicles, neglecting pedestrian safety. Numerous comprehensive studies have explored various facets of pedestrian safety.

## 2. Pedestrian Crossing Warrants

In certain instances, advanced pedestrian crossing infrastructure has been installed where it might not be warranted, while in crucial situations demanding pedestrian crossing facilities, none has been provided. The current criteria guiding the implementation of pedestrian crossing facilities primarily focus on factors such as pedestrian and vehicular volume. These criteria are developed considering convenience, alternative crossing options, acceptable gaps, delays for vehicles and pedestrians, roadway design, and various cost-related factors. However, few specific quantitative and qualitative criteria are used to establish warrants related to pedestrian crossing facilities. These include threshold values, priority ranking systems, economy-based assessments, system-based considerations, policy implications, and political factors [11].

Moore and Older conducted a study evaluating grade-separated pedestrian crossings, introducing a convenience factor 'R', which signifies the ratio of travel time between grade-separated and at-grade crossing facilities. An 'R' factor of 1 indicates equal travel times, prompting pedestrians to favour grade-separated crossings [12]. Axler discussed the considerations for installing overpasses and underpasses, examining macroscopic, geographic, convenience, alternatives, safety, traffic operations, design, and cost factors [11]. The proposed warrants by Axler predominantly encompass quantitative criteria, like pedestrian and vehicular volume and speed, as well as qualitative elements, such as topography, lighting, land use, and funding, for grade-separated pedestrian crossings.

Zegeer et al. sought to identify links between frequently occurring accident types and corresponding mitigation strategies in urban areas. Experts provided mitigation strategies for urban, motorway, and rural roads, classifying various accident types based on their nature and implementing remedial measures [13]. Braun and Rodin quantified the benefits of segregating pedestrian and vehicular traffic, considering safety, social impact, economics, the environment, and health. They identified 36 parameters through extensive research involving transportation agencies, government offices, etc., [14].

The initial development of pedestrian crossing warrants based on the $PV^2$ criteria originated in the U.K., along with a detailed site assessment framework reported by the DfT [15,16]. The type of pedestrian crossing facilities is then identified based on the adjusted $PV^2$ value. Initially, this approach involves plotting graphs correlating pedestrian volume (P) and vehicular volume (V) to establish recommended threshold values [17–22]. India also adopts $PV^2$ criteria-based warrants outlined in IRC-103, 'Guidelines for pedestrian facilities' [23], initially introduced in 1988. Similar to the U.K. warrants, this criterion is widely embraced by planners, government bodies, and non-governmental agencies in India for pedestrian-related planning and provisions [15]. The City of River Falls reported point-based multiple-criteria PCWs as a combination of macroscopic and microscopic factors [24]. Points are assigned on a scale of 10 to 8 h pedestrian volume (macroscopic),

peak hour pedestrian volume (macroscopic), and the average number of accepted gaps in a 5 min period (microscopic).

IRC-103 (2012) generally adheres to the $PV^2$ threshold value concept to determine various pedestrian crossing facilities. However, it lacks specific guidance on whether to provide overpasses, underpasses, skywalks, hump crossings, etc. Instead, it outlines only four conditions for implementing grade-separated pedestrian crossing facilities, as described below [23].

- $PV^2 > 10^8$ for undivided roads or $PV^2 > 2 \times 10^8$ for divided roads.
- Vehicle Speed > 65 kmph.
- Waiting time for pedestrians/vehicles is too long.
- Pedestrian injuries > 5 per year.

Teja introduced pedestrian crossing criteria centred on lane crossing times, tailored explicitly for 4-lane- and 6-lane-divided roads. Their study reported grade-separated pedestrian crossing facility threshold values at 34 s for 4-lane roads and 31 s for 6-lane roads [25]. Prabhu and Sarkar conducted a detailed investigation into pedestrian behaviour, analysing gap acceptance, pedestrian speed, and how pedestrians navigate road crossings. Their study highlighted the significance of pedestrian delays, accepted gaps, and platoon size in decision making during road crossings. They established relationships among variables to determine critical gaps, pedestrian speeds, platoon sizes, and vehicular speeds [26]. In 2007, the New Zealand Transport Agency developed a guideline document focusing on safe pedestrian movements [27]. The guideline uses tables and flowcharts based on pedestrian crossing level-of-service (LoS) criteria based on average pedestrian delay. They recommend the value of delay by pedestrians based on the delay table developed based on the modified Tanner's delay function. These modifications were made to the original delay function [28] and were explained by Abley et al. [29].

The Pedestrian Crossing Control Manual for British Columbia [30] is the guideline established and followed for PCWs in Canada. It provides recommendations on the type of crosswalk to be installed for a given number of crossing opportunities and the equivalent adult unit (EAU) of pedestrians per hour.

Kadali et al. comprehensively analysed pedestrian–vehicle ($PV^2$) conflicts at eight distinct unprotected mid-block crosswalks in Mumbai. Using a K-means cluster algorithm, they established threshold values based on $PV^2$ analysis. This analysis involved grouping crosswalks according to $PV^2$ value adjustment factors. Their findings suggested that a $PV^2$-value of $5 \times 10^8$ or higher necessitates the implementation of a midblock crosswalk featuring zebra markings, a threshold surpassing values proposed by researchers in developed nations [31]. In the context of current traffic conditions, Jain et al. subsequently revised $PV^2$ values. They extracted data on maximum hourly pedestrian flow and critical gap using videography across mid-block sections spanning 2-lane, 4-lane, 6-lane, and 8-lane carriageways. The collected $PV^2$ data exhibited a normal distribution, with a higher priority to vehicular traffic. The assessment of the cumulative distribution frequency of $\log_e (PV^2)$ pinpointed threshold value changes at the 2nd, 5th, and 75th percentiles across all lanes [32]. Additionally, Golakiya et al. proposed warrants based on $PV^2$ threshold values for 4-lane- and 6-lane-divided roads. Their study estimated the impact of pedestrian crossings on mid-block capacity under mixed traffic conditions, concluding that midblock capacity remains unaffected up to pedestrian flows of 200 pedestrians per hour [33].

## 3. Data Collection and Extraction

Before selecting the primary study area, a pilot survey was conducted at 12 distinct locations, with varying lane configurations, pedestrian facilities, and surrounding characteristics. The key criteria guiding the selection of the primary study area included the maximum hourly vehicular flow (V) and the maximum hourly pedestrian volume (P), which establish the upper limits of the $PV^2$ matrix. The sole site meeting these selection criteria was designated as the primary study area among the selected sites in Ahmedabad. The primary site is an unsignalised intersection, featuring a 6-lane primary stream (excluding a

two-lane BRTS corridor) and a 4-lane minor stream at Isanpur Crossroad. The other site is the CTM Expressway junction. Both locations, situated along national highways, traverse densely populated urban areas. Isanpur Crossroad is an unsignalised intersection, and the CTM expressway junction was signalised. Further, at the Isanpur Crossroad, minimal pedestrian crossing facilities are provided, while at the CTM expressway junction, two overpasses are supplied for pedestrians to cross the road. Thus, the second site is selected for validation purposes only. An extensive analysis of the existing literature identified the essential primary data necessary for developing a model and establishing standards for appropriate pedestrian crossing facilities. These parameters include vehicular volume, pedestrian volume, accepted pedestrian gap, vehicular density, and pedestrian waiting times. The construction of the $PV^2$ threshold value chart was based on peak hour vehicular (V) and pedestrian flow (P). Integral to the model development were considerations of gap acceptance and pedestrian waiting times in seconds aligned with peak-hour vehicular flow.

The on-site collection of primary data utilised videography techniques [32]. This involved conducting a six-hour videography survey in two-hour intervals, from 6:30 to 8:30, 10:00 to 12:00, and 16:30 to 18:30, to ascertain the peak hour. The video playback was slowed to 0.0625-times the actual speed, meticulously capturing pedestrian and vehicular movements. Vehicles were categorised into distinct classes—such as two-wheelers, three-wheelers, four-wheelers, trucks, buses, non-motorised vehicles, animal-driven vehicles, and specialised vehicles, like cranes or rollers, with pickup or light motorised vehicles further classified under three-wheelers or four-wheelers.

Classified vehicular volume data were gathered at every 5 min interval throughout the six hours (06:31–06:35, 06:36–06:40, 06:41–06:45, and so on); the density data were derived from vehicle class counts by pausing the video at every 30 s interval for a total 6 h. Regardless of how they crossed the road, pedestrian movements were documented concurrently with vehicular observations. The primary site for data collection is shown in Figure 1.

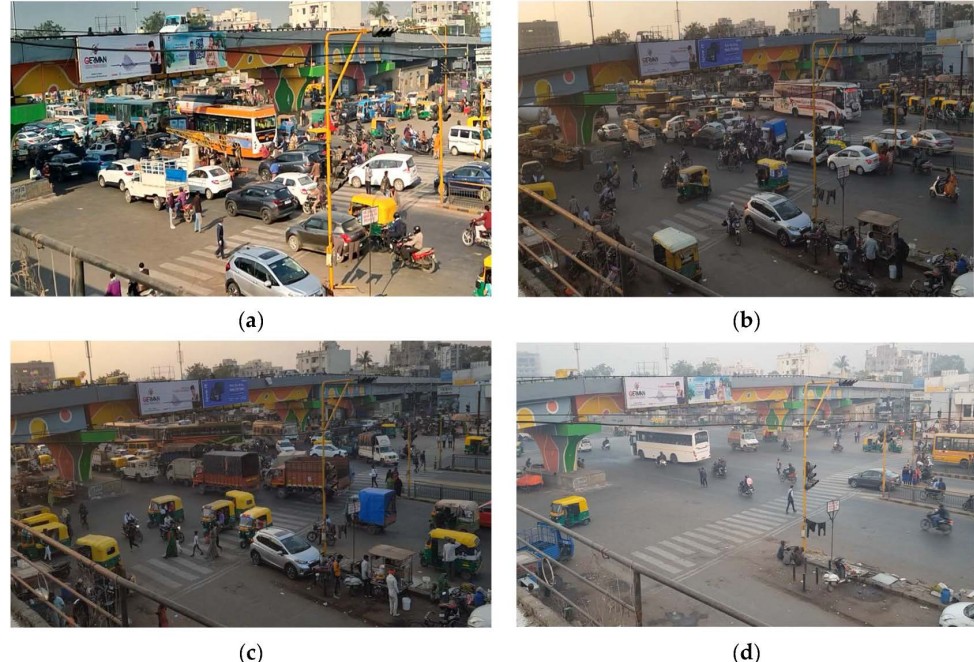

**Figure 1.** Photographs of the primary site (Isanpur Crossroad) for data collection at the (**a**) afternoon peak, (**b**,**c**) evening peak, and (**d**) early morning peak.

Microscopic parameters, such as the gap accepted by pedestrians and pedestrian delay, were used in the former guidelines of countries, such as Canada [30], New Zealand, [27], and the UK [16]. A pedestrian crossing decision is usually based on a safe gap, which is perceived differently by each individual [34]. Thus, gap acceptance between the vehicles in

seconds was also considered in this research. Gap acceptance data were extracted manually [35] from the video recordings. The data extraction concerning pedestrian-accepted gaps was conducted explicitly during peak hours, specifically from 17:30 to 18:30. This time frame coincides with high volumes of both pedestrians and vehicles, replicating conditions where pedestrians face the most significant challenges in crossing a 6-lane unsignalised carriageway lacking designated pedestrian facilities. The critical gap is minimised in scenarios marked by elevated pedestrian and vehicular volumes, underscoring the demanding conditions of highly diverse and random vehicular movements intersecting with unpredictable pedestrian behaviour. A total of 929 instances of gap acceptance data were extracted from 1037 pedestrians observed. On such wide carriageways, pedestrians mostly accept one or more gaps between the vehicles. Hence, the carriageway was divided into three parts for pedestrians crossing, from the kerb on one end to the kerb on the other. Pedestrians coming from BRTS station accept only one gap on only one lane of the carriageway, and because of this, a large number of gap-accepted data were extracted. The carriageway was divided into three parts, as shown in Figure 2. Those coming from the BRTS stand must cross either carriageway lane.

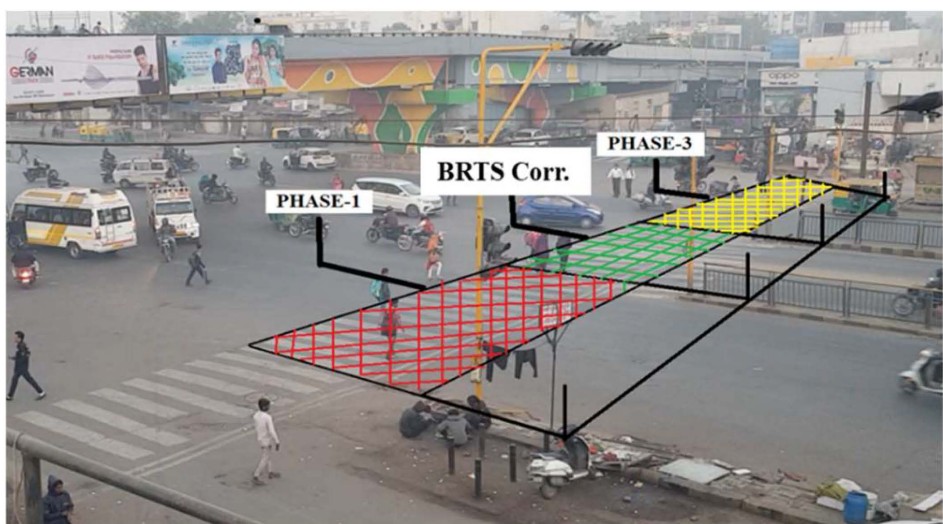

**Figure 2.** Gap acceptance phase representation on actual site.

The manual extraction of gap acceptance data followed a specific procedure. When a pedestrian approached the carriageway kerb intending to cross, their arrival time was recorded. Simultaneously, the arrival time of vehicles at the intersection or crossing path was noted in the dataset. For instance, if a pair of pedestrians arrived to cross at 18:14:19 h and a vehicle crossed the same path after a 2 s interval, followed by another vehicle in close succession, resulting in a 2 s gap that the pedestrians did not use, it was not recorded. However, if a subsequent vehicle crossed the path after a 6 s interval while the pedestrians managed to cross within that time, this 6 s gap was considered acceptable by that group of pedestrians. This procedure was applied across each crossing phase, determining the type of gap accepted by pedestrians in seconds, characterised as a rolling gap.

The waiting duration of pedestrians can be derived from the gap acceptance data structure. Specifically, the pedestrian waiting time represents the interval from when pedestrians reach the carriageway kerb, awaiting an appropriate gap to cross, until they identify and use a suitable opening. This waiting period varies; the waiting time may be null if pedestrians identify either no vehicles or vehicles at a satisfactory distance. Throughout this research, the maximum waiting time observed for pedestrians was 63 s. This waiting period is significantly influenced by both vehicular volume and pedestrian behaviour. For instance, elderly pedestrians tend to wait longer for more significant gaps to cross the carriageway than younger pedestrians.

## 4. Data Analysis

### 4.1. Carriageway Capacity

The data analysis reveals a predominant presence of two-wheelers (2-W) on the carriageway, constituting a significant portion of vehicular traffic. Specifically, the primary composition of vehicular traffic comprised approximately 30% 2-W, 17% 3-W, 22% 4-W, and 11% trucks and buses combined.

As per Indo-HCM 2017, capacity at an unsignalised intersection is defined for each non-priority movement or stream [36]. The capacity of a four-legged unsignalised intersection with a 6-lane major and 4-lane minor carriageway was estimated by referring to Indo-HCM 2017 [36] after applying the adjustment factor over the base condition. Classified and directional traffic volume counts are extracted and converted into PCU. Conflicting flow was estimated using the methodology suggested in Indo-HCM 2017. Another step for capacity estimation is based on the critical gap values for the movement based on the occupancy time method. Site-specific critical gaps for different movements can be obtained using the base value and then adjusting them for the proportion of heavy vehicles in the conflicting traffic streams. So, the critical gap for any movement can be obtained using Equation (1) below [36].

$$t_{c,\,x} = t_{c,\,base} + f_{LV} \times ln(P_{LV}) \tag{1}$$

The capacity calculation involved Equation (1), considering the base critical gap, adjustment factors for vehicle types, and the proportion of heavy vehicles in conflicting traffic streams. Equation (2) was employed to determine the capacity of each movement, resulting in an estimated capacity of 10,600 PCU/h for the unsignalised 4-legged intersection, as per the methodology outlined in Indo-HCM 2017.

$$C_x = a \times V_{c,\,x} \frac{e^{\frac{-V_{c,x}(t_{c,x}-b)}{3600}}}{\left(1 - e^{\frac{-V_{c,x} \times t_{f,x}}{3600}}\right)} \tag{2}$$

$C_x$ = capacity of movement 'x' (in PCU/h), $V_{c,x}$ = conflicting flow rate corresponding to movement x (PCU/h), $t_{c,x}$ = critical gap of standard passenger cars for movement 'x' (s), $t_{f,x}$ = follow-up time for movement 'x' (s), and 'a' and 'b' = adjustment factors based on intersection geometry [36].

To refine this estimation, adjustments were made by including the capacity of through traffic on the major road, treated as a mid-block, and evaluated using the Greenshields model. This involved considering vehicular volume, density, and speed about fundamental traffic flow parameters. Consequently, the cumulative estimated capacity for unsignalised intersections with a 6-lane major stream reached 11,300 PCU/h. This method, albeit approximate, closely mirrors the capacity of a 6-lane mid-block.

Simultaneously, pedestrian counts were conducted alongside vehicular volume analysis. The maximum number of pedestrians observed within the six-hour dataset, particularly during the peak flow period, amounted to 1399. These peak values of pedestrian (P) and vehicular (V) counts were taken as the maximum values in the $PV^2$-matrix.

### 4.2. $PV^2$ Analysis

The obtained values of 'P' and 'V' were utilised to establish the range within the $PV^2$-matrix. Starting from zero, the 'P' and 'V' values were incremented by 100 in both directions. The maximum values of 'P' and 'V' represented the carriageway capacity and the peak pedestrian flow. This process allowed for deriving all potential $PV^2$ values, encompassing every conceivable combination of 'P' and 'V'. To streamline further analysis, the $PV^2$ values were converted into logarithmic base ten values to s, generating an additional $log_{10}(PV^2)$ matrix. Following data classification, a frequency distribution table was compiled.

A Kolmogorov–Smirnov (KS) streamlined further analysis test was initially conducted on the $log_{10}(PV^2)$ values, revealing a heavily negatively skewed distribution rather than a

normal distribution. Further, the Kolmogorov–Smirnov test was performed on the selected data, indicating a possible $\log_{10}$ ($PV^2$) value on such a carriageway, indicating that data are normally distributed [32]. Interestingly, upon analysing this subset of data, it was found to follow a normal distribution. The KS test was performed using the Lilliefors test, using the Lilliefors distribution table instead of the KS test table values to validate the obtained result. This test, designed for assessing normality, utilises Lilliefors tables, characterised by smaller critical values compared to other normality tests, thereby reducing the likelihood of identifying data as normally distributed. Despite this characteristic, the Kolmogorov–Smirnov (KS) test conducted on the selected data affirmed the normal distribution. Furthermore, to enhance the robustness of this conclusion, a supplementary Lilliefors test was conducted, providing additional validation of the dataset's normality. This comprehensive approach bolsters confidence in the normal distribution of the data, thereby enabling more precise statistical analysis and interpretation of results.

The result of the Lilliefors test indicates that there is no significant difference from the normal distribution (D (32) = 0.14 and *p*-value = 0.12). Here, a *p*-value of 0.1176 and a D value of 0.1394 at a 95% confidence level ($\alpha$ = 0.05) are obtained, and since the *p*-value > $\alpha$, we accept the $H_0$ (null hypothesis), i.e., the data are assumed to be normally distributed. The Kolmogorov–Smirnov test was also performed in MS Excel 2016 since the static difference between the actual theoretical values obtained is 0.1149, which is less than the critical value for n = 32, which is 0.24008, obtained from the Kolmogorov–Smirnov table. Since D (D+ or D−) < *p*-value, the data extracted are normally distributed. The Q-Q plots are shown in Figure 3.

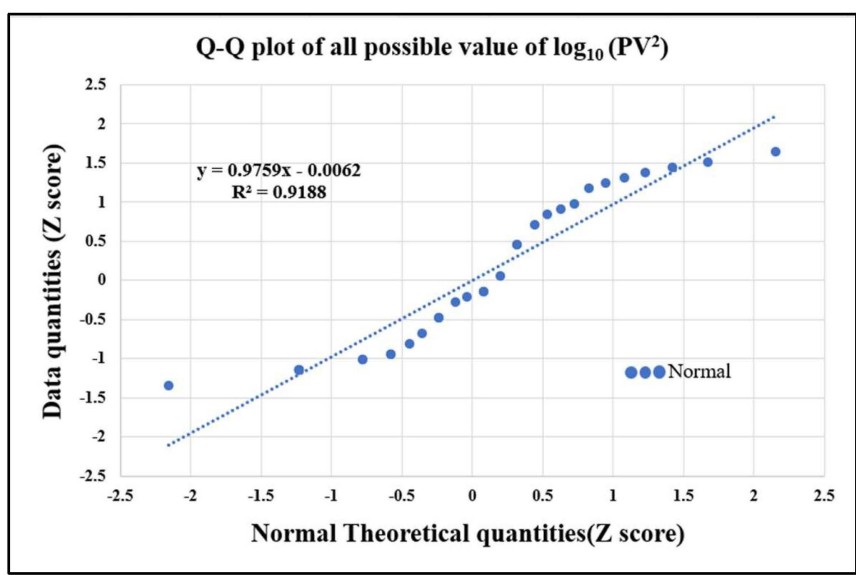

**Figure 3.** Q-Q plot of selected value of pedestrian volume and vehicular volume obtained in MS Excel 2016.

The threshold value parameter was identified primarily based on cumulative frequency distribution graphs, aligning with the 15th, 50th, and 85th percentiles, in line with previous research recommendations [31,32,37]. Due to the negatively skewed distribution, the corresponding percentage values for the change in curvature deviated from the usual norm. The observed percentage values were 8%, 44%, and 88%, correlating to $\log_{10}(PV^2)$ values of 8.65, 10.08, and 10.90, respectively, translating to $PV^2$ values of $4.47 \times 10^{08}$, $1.20 \times 10^{10}$, and $7.95 \times 10^{10}$. Figures illustrating the cumulative frequency distribution graph with log10($PV^2$) corresponding to the percentage values are depicted in Figure 4.

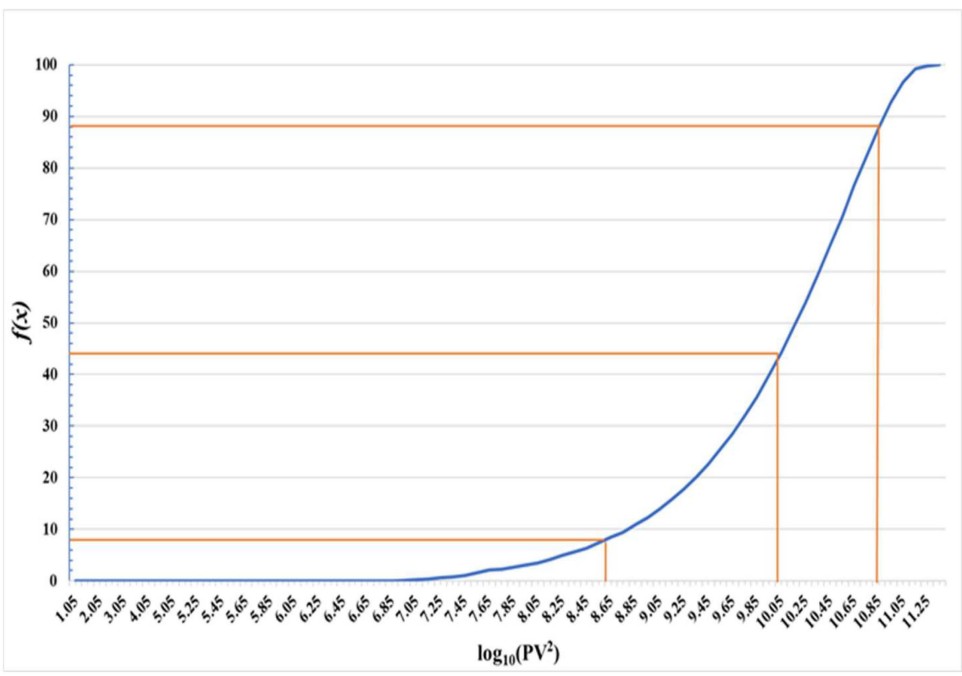

**Figure 4.** Percentage cumulative frequency distribution curve for Isanpur site of 6-lane 2-way carriageway considering V as the vehicular capacity of the carriageway.

### 4.3. Gap Acceptance and Waiting Time Analysis

The gap data were extracted manually [35]. The dataset obtained encompasses 7273 gaps observed across 1039 instances of pedestrians crossing the carriageway, either individually or as a group. Thus, 1399 pedestrians in all were noted during the gap acceptance data collection process. Within the 7273-gap dataset, 1125 gaps were acknowledged as accepted, while 6148 were rejected. Among the accepted gaps, 196 instances were zero duration gaps. This occurrence might appear peculiar, yet it is plausible. For instance, when an individual reaches the kerb simultaneously with approaching vehicles, they might pause briefly before starting to cross, managing to reach a section of the carriageway but not completing the entire crossing due to oncoming traffic. In such scenarios, this is considered a zero-second accepted gap if pedestrians halt midway or vehicles pass in proximity without impeding the person's movement, resulting in no elapsed time for the gap. These zero accepted gaps were excluded from the subsequent analysis [35]. The critical gap represents the shortest duration in seconds that pedestrians are willing to accept to safely cross a carriageway under prevailing traffic conditions.

The methodology involved categorising the data into multiple ranges, from 0–1 and 1.01–2 to 2.01–3, and so forth, up to the maximum accepted gap value. A table tabulating the number of gaps falling within each range was generated. Cumulative values for gap classes were calculated, and corresponding columns noted that the count of gaps did not fall within those ranges. A graph was plotted to depict the relationship between accepted gaps within each class and the count of non-accepted gaps within the same class. In this instance, the critical gap was identified as the value where both curves intersected, determined to be 2.05 s. The graph is shown in Figure 5.

A total of 850 waiting times were extracted. With an increase in vehicular volume, the available gap will decrease, and on the opposite side, the waiting time for pedestrians will increase. Here, pedestrian gap acceptance and waiting time analysis are not performed when combined with vehicular volume.

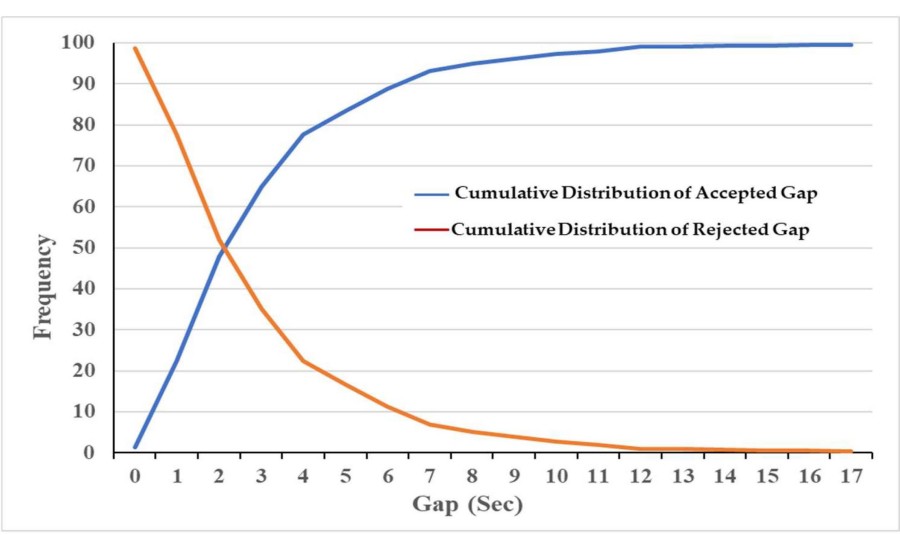

**Figure 5.** Critical gap obtained by Raff's method.

*4.4. Level-of-Severity (LOSe) Analysis*

In this study, the clustering method is used for the level-of-severity analysis. The K-means clustering technique was used and performed in IBM-SPSS 2019. The dataset contains vehicular volume, vehicular density, and gap acceptance by pedestrians, of which three datasets are obtained and analysed individually. The K-clustering technique is used, as the best algorithm found is K-mean clustering. Finding similar clusters takes less time than other clustering algorithms [38]. K-means clustering is a method of vector quantisation, originally from signal processing, that aims to partition n observations into k clusters, in which each observation belongs to the cluster with the nearest mean (cluster centres), serving as a prototype of the cluster, which means that the value has some similarities with all other values in the cluster. This technique is used to classify given data objects into different k clusters through the iterative method, which tends to converge to a local minimum. So, the outcomes of the generated clusters are dense and independent. IBM SPSS performed K-means clustering. The drawback of this method is that it does not specify the optimum number of clusters for a specific data type.

Before clustering, it is necessary to determine the optimum number of clusters; the elbow curve method was used to decide this. It is a well-known method, in which the sum of squares at each number of clusters is calculated and graphed, and where the curve is smooth after the steep slope; a point of change in slope indicates the optimal number of clusters in further cluster analysis. This procedure is inexact but still helpful. Weka 2021 software is used to analyse the sum of squared errors. From this analysis, the graph in Figure 6 clearly shows that the optimum number of clusters based on the data was three. As only three optimum clusters were effective for analysis based on the three clusters, four levels of severity were assigned, namely high risk, medium risk, low risk, and very low risk. The severity level was decided based on the facility assigned at different stages of the $PV^2$ threshold values, which is further based on the level of the interaction of the vehicle and pedestrians, which is explained in Section 5.1. The final range of the level of severity is obtained, as shown in Table 1.

**Table 1.** Range of every parameter corresponding to risk level.

| Class | Vehicular Volume (PCU/h) | Vehicular Density | Gap Acceptancy (Sec) | Waiting Time (Sec) |
|---|---|---|---|---|
| High Risk | >8665 | >162 | <2.55 | >33 |
| Medium Risk | 8665–7334 | 162–120 | 2.55–3.47 | 33–13 |
| Low Risk | 7334–5740 | 120–90 | 3.47–4.63 | 13–04 |
| Very Low Risk | <5740 | <90 | >4.63 | <4 |

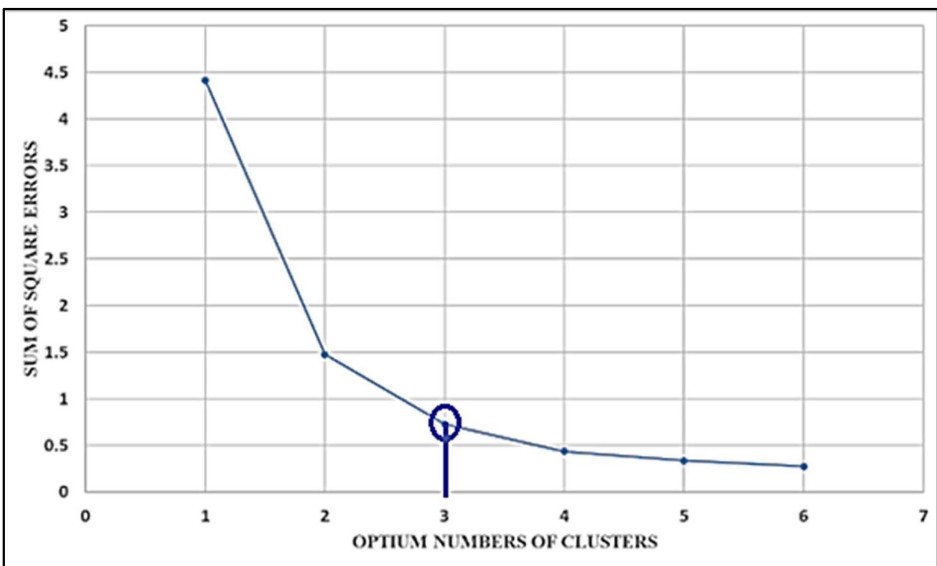

**Figure 6.** Elbow curve showing an optimum number of clusters.

### 4.5. Vehicular Flow vs. Gap Acceptancy

Several approaches were taken to develop a more precise model. Initially, graphs were prepared and analysed using normal and logarithmic values for gap acceptance with linear, parabolic, and logarithmic relations with vehicular volume. This led to the development of two models for peak-hour vehicular flow and gap acceptance, one using normal gap acceptance values and the other employing logarithmic values, demonstrating a parabolic relationship.

Two models were considered from these evaluations, and only the most effective two will be further explored.

In a separate graph, the y-axis represents vehicular volume in PCU/h, while the x-axis represents the log gap accepted by pedestrians with a parabolic relation, displaying an $R^2$ value of 0.6213, as shown in Figure 7. The volume of vehicles increases in tandem with a corresponding increase in gap size. However, this trend reaches a limit, typically when vehicular volume peaks. At this point, pedestrians tend to accept significantly larger gaps, creating a steep incline in the graph. Beyond this peak, the pedestrian acceptance of gaps increases as vehicular volume declines moderately.

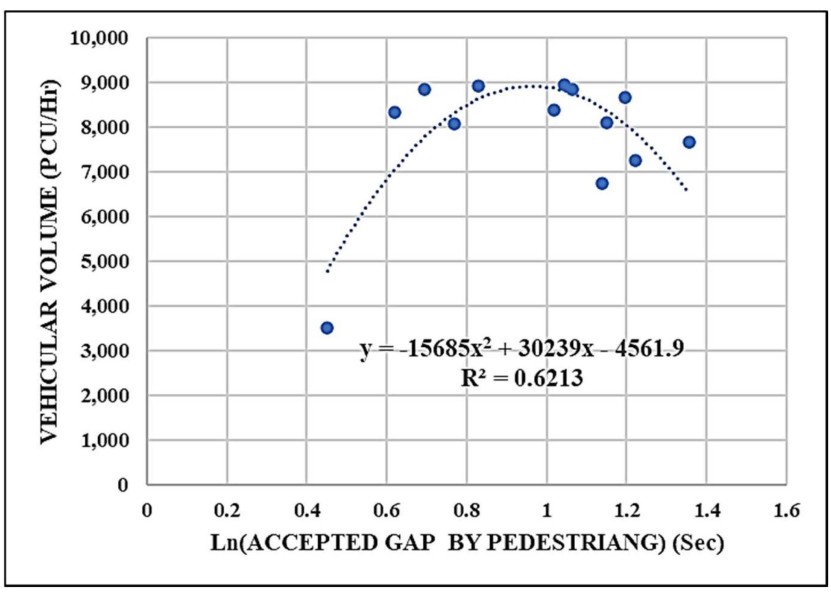

**Figure 7.** Vehicular volume vs. ln (gap acceptancy) with parabolic relation.

### 4.6. Waiting Time Relation Analysis

A similar methodology explored the relationship between vehicular volume and waiting time. Identical datasets were chosen for this analysis. A logarithmic relationship between both parameters was selected due to practical conditions; this implies that as vehicular volume increases, delay also consistently escalates. The resulting graph is presented below.

In the depicted graph, the y-axis illustrates vehicular volume in km/h, while the x-axis represents pedestrians' waiting time, showcasing a logarithmic relationship with an $R^2$ value of 0.6271, as shown in Figure 8. There is a rapid increase in waiting time up to the peak vehicular volume, after which the increment occurs more slowly. A brief explanation of the relationship is given in Section 5.2.

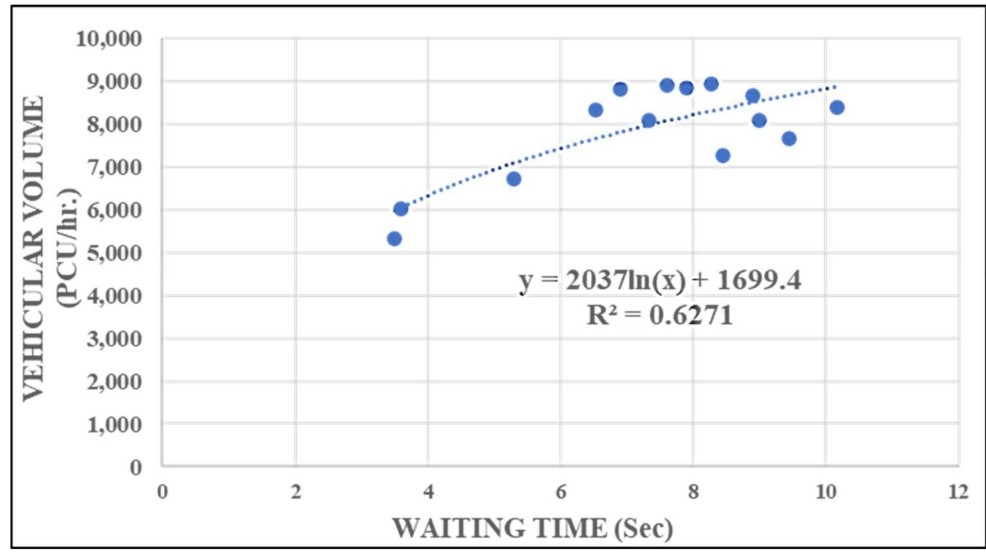

**Figure 8.** Graph of vehicular volume vs. waiting time with logarithmic relation.

## 5. Results and Interpretation

### 5.1. $PV^2$-Based Pedestrian Crossing Warrant

Pedestrian crossing warrants are established by evaluating the standard values associated with percentage shifts in the cumulative frequency distribution curve (CFD). These percentage values, observed at 8%, 44%, and 88% on the CFD, align with the peak-hour vehicular volume. Four distinct pedestrian crossing facility types are determined based on these values: grade-separated crossings, signalised zebra crossings (with pedestrian crossing time or manual control during peak hours), and minimal or no crossing facility. These categories, designated as Stage-1 (<8%), Stage-2 (8–44%), Stage-3 (44–88%), and Stage-4 (>88%) based on peak-hour vehicular flow, reflect the increasing challenges pedestrians face as vehicular volume rises. The delineation into these stages is informed by the dynamics between vehicles and pedestrians, emphasising the imperative for heightened pedestrian safety. Locations exhibiting higher $PV^2$ values necessitate more advanced crossing facilities. The P vs. V chart illustrating this relationship is depicted in Figure 9, while the corresponding recommended crossing facilities are detailed in Table 2.

**Table 2.** Pedestrian crossing facility corresponding to a range of $PV^2$ for vehicular capacity.

| Sr. No. | $\log_{10}(PV^2)$ | $PV^2$ Range | Pedestrian Crossing Facility |
|---|---|---|---|
| 1 | <8.65 | <$4.47 \times 10^{08}$ | Nominal crossing facility |
| 2 | 8.65–10.08 | $4.47 \times 10^{08}$–$1.20 \times 10^{10}$ | Manually controlled zebra crossing |
| 3 | 10.08–10.90 | $1.20 \times 10^{10}$–$7.95 \times 10^{10}$ | Signalised zebra crossing |
| 4 | >10.90 | >$7.95 \times 10^{10}$ | Grade separated crossing |

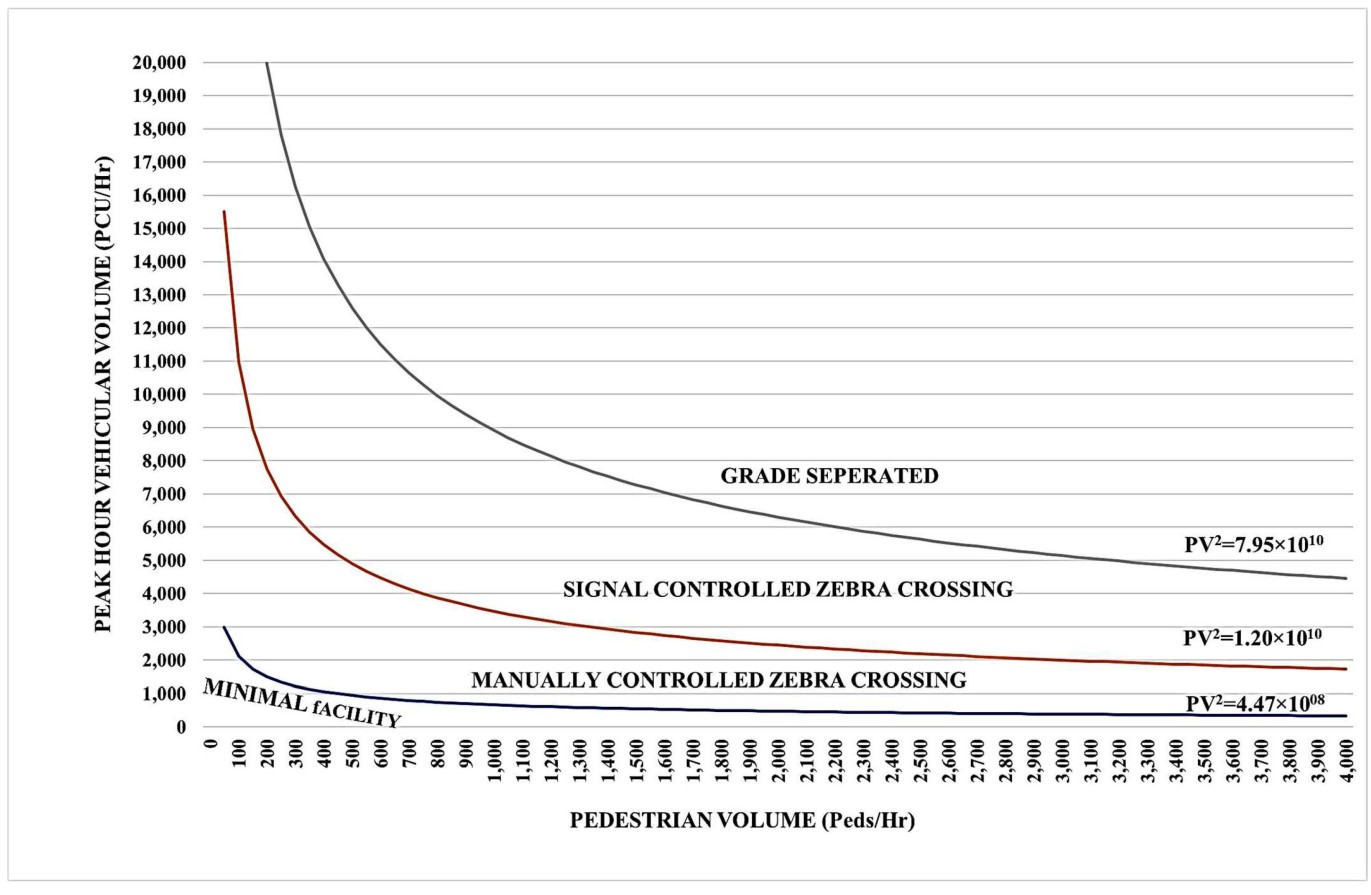

**Figure 9.** $PV^2$ threshold value warrant chart.

*5.2. Gap Acceptance and Waiting Time-Based Pedestrian Crossing Warrants*

Pedestrian crossing warrants are formulated based on the correlation between vehicular volume and gap acceptance. As discussed, higher severity levels necessitate safer pedestrian facilities. Utilising the K-means clustering method, three centres are derived for each parameter. These values are pivotal in determining the severity level and the corresponding requisite pedestrian crossing facilities. As previously explained, the three centres enable the simulation of four conditions: providing grade-separated crossings, signalised zebra crossings, manually controlled zebra crossings, or no facility at all.

As shown in Figure 10, the volume of vehicles increases in tandem with a corresponding increase in gap size. However, this trend reaches a limit, typically when vehicular volume peaks. At this point, pedestrians tend to accept significantly larger gaps, creating a steep incline in the graph. Beyond this peak, the pedestrian acceptance of gaps increases as vehicular volume declines moderately. Initially, when vehicular volume is low, the accepted gap size is also limited due to the analysis methodology and the unpredictable nature of traffic flow. Density data served as the primary dataset, with vehicular volume estimated through curve fitting techniques. Real-world conditions reveal fluctuating traffic flow during peak hours, leading to corresponding fluctuations in accepted gap values—lower acceptance for higher volumes, and vice versa. Hence, average vehicle volume is utilised for deeper analysis. In scenarios of low vehicular volume, gaps tend to be narrower due to rapid and uncertain volume changes over short time intervals. Although such cases are rare in the dataset, they are plausible, as evidenced by the ascending portion of the curve depicted in Figure 10. However, these occurrences were infrequent within the dataset. The descending portion of the curve was considered pivotal for warrant development.

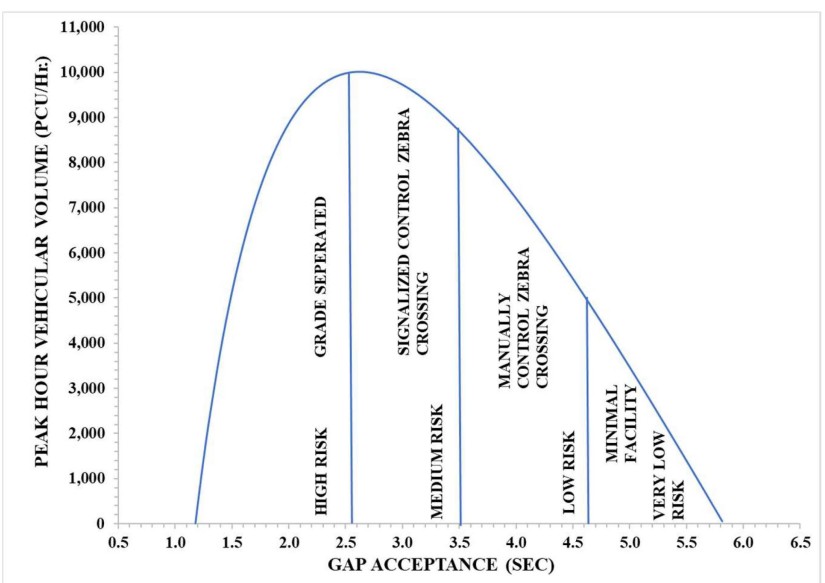

**Figure 10.** Warrant chart representing the severity region and corresponding pedestrian crossing facility considering gap acceptance parameter.

The relation between waiting time by pedestrian and vehicular volume is represented in Figure 11. There is a rapid increase in waiting time up to the peak vehicular volume, after which the increment occurs more slowly. Pedestrians typically remain on the kerb until a certain threshold of vehicular volume is reached. However, once this limit is surpassed (typically during peak traffic hours), pedestrians begin to take the risk of crossing the road, despite the ongoing flow of vehicles. They persistently move onto the road to cross, regardless of the traffic volume in the lane. This behaviour prompts vehicle drivers to adjust their driving, often manoeuvring closer to the side of the road. As a result, pedestrians consistently accept smaller gaps and are less inclined to wait, which is also influenced by platoon behaviour, affecting vehicle drivers' behaviour.

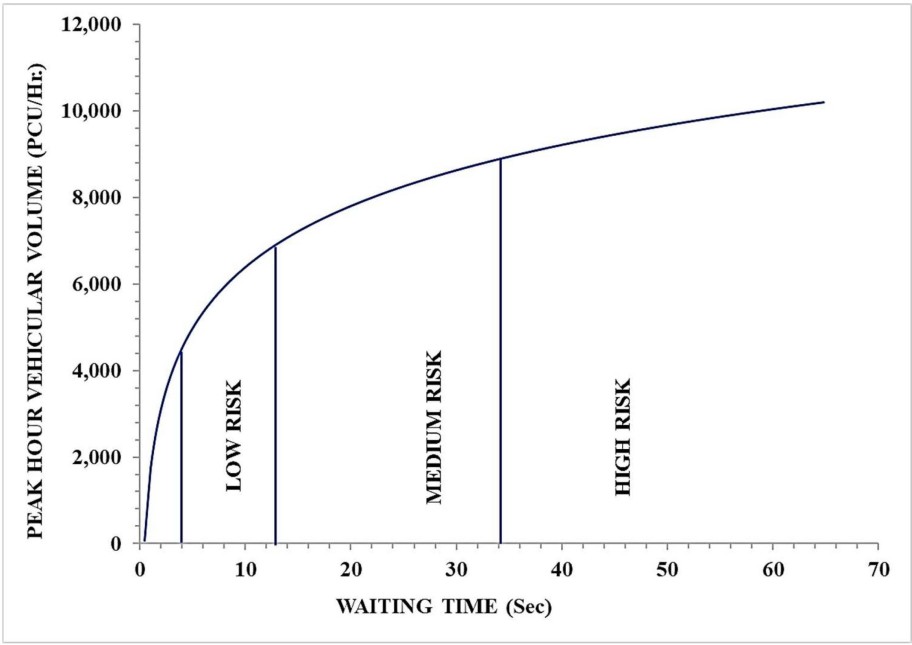

**Figure 11.** Warrant chart representing the severity region and corresponding pedestrian crossing facility considering waiting time by pedestrian as a parameter.

Models depicting waiting time and gap acceptance concerning peak-hour vehicular flow are illustrated in Figures 10 and 11 to guide the determination of pedestrian crossing facilities.

*5.3. Proposed Pedestrian Crossing Warrants*

- $PV^2$ chart, where V is peak hour vehicular flow and P is peak hour pedestrian flow, gap acceptance (GA) in seconds, and waiting time (WT) in seconds.
- If $PV^2 > 7.95 \times 10^{10}$, GA by the pedestrians is <2.55 s and WT by pedestrians is >33 s, then no facility other than a grade-separated pedestrian crossing facility should be provided as there is high vehicle and pedestrian interaction.
- If $1.20 \times 10^{10} < PV^2 < 7.95 \times 10^{10}$, gap acceptance by the pedestrians is 2.55 s < GA < 3.50 s and waiting time by the pedestrians is 13 s < WT < 33 s, then pedestrians are at medium risk, and a signal-controlled zebra crossing with pedestrian signal time should be provided.
- If $4.47 \times 10^8 < PV^2 < 1.20 \times 10^{10}$, gap acceptance time by pedestrians is 3.50 s < GA < 4.65 s and waiting time by the pedestrians is 4 s < WT < 13 s, then manually controlled zebra crossings during peak hours should be deployed.
- If $PV^2 < 4.47 \times 10^8$, gap acceptance by the pedestrians is >4.65 s and waiting time by the pedestrians is <4 s, then no facility or a minimal pedestrian crossing facility is required at the site.

## 6. Conclusions and Limitations

The unsignalized intersection's capacity reached 11,300 Passenger Car Units per hour (PCU/h) following adjustments made to the Indo-HCM 2017 methodology. Comparing this capacity with the same lane in the mid-block section revealed a slight reduction. The identified $PV^2$ threshold values, ranging from $4.47 \times 10^{08}$ to $7.95 \times 10^{10}$, exceeded the proposed values in various Indian practices, such as IRC:103-(1988, 2012) and UK warrants (1987). Pedestrians' accepted gap was measured at 2.05 s, notably less than other intersections, such as the 4.125 s observed by Yohannes et al. (2019) [39]. This discrepancy arose due to the lower speed of incoming vehicles compared to the mid-block section, where a free flow speed of 12.3 kmph was recorded. Consequently, pedestrians accepted narrower gaps, feeling safer despite significantly high vehicular volume. Pedestrian volume here exceeded the one observed by Yohannes et al. (2019) by 85%. In peak-hour conditions, both vehicular flow and accepted gap values exhibited fluctuations, resulting in lower accepted gaps at higher volumes or vice versa. Therefore, average vehicular volumes were used for further analysis, accounting for potential instances of lower accepted gaps amidst unpredictable vehicular volume fluctuations.

At the Isanpur location, vehicular capacity and peak-hour vehicular flow stood at 11,400 PCU/h and 9800 PCU/h, respectively, with a peak-hour pedestrian volume of 1399 Peds/h. $PV^2$ analysis recommended a grade-separated pedestrian crossing facility as it exceeded the curve of $7.95 \times 10^{10}$. Both gap acceptancy and critical gap analysis indicated a high risk, signifying increased vehicle–pedestrian interactions, further advocating for a grade-separated pedestrian crossing. Similarly, at the CTM expressway junction, with a peak-hour vehicular flow of 710 PCU/h and peak-hour pedestrian flow of 548 Peds/h, the gap and waiting time fell within the low- and medium-risk categories, signalling no necessity for additional overpasses as crossing facilities.

Regarding the applicability of the warrants proposed in this study, they are grounded on the $PV^2$ relationship, acknowledging acceptable gaps and waiting times for pedestrians. Vehicular volume (V) is quantified in Passenger Car Units (PCU) per hour. Any alteration in the composition of vehicles will directly impact vehicular volume. A modification in V results in a corresponding adjustment in $PV^2$, indicating the appropriate crossing facility for the intersection. Despite being a six-lane major stream lacking traffic signals, the vehicular volume may vary, influencing the choice of appropriate facilities, as observed in this study after analysing sites with similar lane configurations and characteristics. Also, a reduction in vehicular volume leads to an increase in average vehicle speed, making

it challenging for pedestrians to identify suitable crossing opportunities. Consequently, pedestrians experience prolonged waiting times at the kerb side, and vice versa. Thus, the warrants are applicable to locations with such characteristics as the selected primary survey site.

However, this study exhibits certain limitations. Although it provides conclusive metrics, such as gap acceptance, waiting time, and critical gap acceptancy, to inform decisions regarding the provision of pedestrian crossing facilities, additional microscopic parameters require identification. The integration of these parameters would further enhance the implementation of the warrants. One of the parameters is human behaviour, which tends to be unpredictable and subjective. Behavioural analysis, such as classification based on criteria, like gender, age, profession, eye contact, or the directional flow of pedestrians, was not conducted.

Creating eye contact with the driver not only enhances pedestrians' positive feelings but also contributes to safer crossing experiences. However, effectively monitoring the driver's line of sight from their perspective presents a notable challenge, particularly when approaching intersections, as evidenced by observations. In such scenarios, drivers must actively scan all incoming traffic directions while remaining attentive to pedestrians crossing the road. It has been observed that riders of two-wheeled vehicles tend to undertake more overtaking manoeuvres compared to drivers of other vehicle types. Additionally, pedestrians face significant risk when encountering a two-wheeled vehicle at intersections. This pedestrian behaviour may stem from the perception that, in the worst-case scenario, they would sustain less injury or damage compared to other vehicles. Furthermore, it has been noted that pedestrians often cross roads incrementally, highlighting the influence of pedestrian experience and their trust in autonomous vehicle drivers. Given time and resource constraints, further investigation into this matter is warranted.

Additionally, driver behaviour is also influenced by the design and efficacy of pedestrian crossing infrastructure. The presence of raised pedestrian crossings notably reduces the frequency of pedestrians approaching crossings. Implementing raised zebra crossings leads to a marked increase in drivers' attention towards pedestrian crossings. In the studied area, only marked zebra crossings were present, posing a challenge to drivers' focus as they approach intersections. Drivers must diligently monitor surrounding traffic, making it challenging to prioritise attention on pedestrian crossings. Pedestrians are typically noticed by drivers only when they are actively crossing the roadway. Various factors, such as pedestrian and driver age, influence their behaviour. For instance, in areas with elderly pedestrians, drivers often accelerate to pass before the pedestrian, whereas with younger pedestrians, this behaviour contrasts. This phenomenon warrants further investigation. Moreover, this study identifies competitive behaviour among pedestrians when crossing roads, leading drivers to adjust their speeds accordingly. Thus, there exists an interdependency between the behaviour of pedestrians and drivers to some extent, necessitating further research to identify and validate additional parameters. A comprehensive understanding of these behavioural aspects is essential for refining pedestrian crossing warrants.

Moreover, pedestrians tend to find alternative methods to navigate heavy traffic, some opting to cross without using overpasses or underpasses due to the effort and time required. These complexities underline the multifaceted nature of pedestrian behaviour and preferences, which this study did not explore extensively.

## 7. Recommendations

Indeed, expanding the scope to encompass various types of carriageways, including signalised and unsignalised intersections, different lane configurations, divided and undivided roadways, and one-way or two-way streets, could yield valuable insights. While the current study focused on a specific intersection due to resource limitations, its findings may be extrapolated to intersections sharing similar characteristics. This approach enables broader applicability of the results across various intersection types, paving the way for a

more comprehensive understanding and potentially enhancing road safety and efficiency in diverse traffic environments.

Additional parameters need consideration within this study, specifically the analysis of pedestrian behaviour and attitudes towards pedestrian crossings, which will offer further conclusive conditions for determining appropriate pedestrian crossing facilities. It is imperative to include economic analyses as part of the research scope. Similar studies should be conducted within mid-block sections. The level of service (LoS) at unsignalised crossroads must be determined, regardless of incoming and outgoing vehicle speeds. To address the limitations of conventional overpass and underpass designs, a novel facility that combines both advantages while eliminating their shortcomings can be explored. This innovative concept involves slightly elevating the carriageway to align the overpass at the carriageway level, creating what can be termed a "partial underpass". Such a design would alleviate the need for pedestrians to navigate ascents or descents and address concerns regarding ventilation, drainage, safety, lighting, and vehicular–pedestrian interactions. This configuration allows vehicles to navigate the carriageway while pedestrians can cross smoothly without hindrance.

Further research should explore the feasibility of this proposed crossing facility. Assessing its impact on vehicular and pedestrian flows and analysing pedestrian behaviour and the comprehensive economic aspects of such grade-separated pedestrian crossing facilities will contribute to developing and enhancing our nation's future infrastructure.

**Author Contributions:** Conceptualization, S.C. and S.D.; Methodology, S.D. and J.S.; Software, S.C. and J.S.; Validation, J.S.; Formal analysis, S.C.; Investigation, J.S.; Resources, A.K.; Data curation, S.C.; Writing—original draft, S.C.; Writing—review & editing, S.D., J.S. and A.K.; Visualization, J.S.; Supervision, S.D. and J.S.; Project administration, S.D. All authors have read and agreed to the published version of the manuscript.

**Funding:** This research received no external funding.

**Institutional Review Board Statement:** Not Applicable.

**Informed Consent Statement:** Not Applicable.

**Data Availability Statement:** The datasets presented in this article are not readily available because they are part of an ongoing study or due to technical/time limitations.

**Conflicts of Interest:** Author Ashu Kedia was employed by the company Urban Connection Limited. However, the research was done in Ashu Kedia's personal time and has nothing to do with Urban Connection Limited. The remaining authors declare that the research was conducted in the absence of any commercial or financial relationships that could be construed as a potential conflict of interest.

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
