# Peer review of "Assessing Traffic Characteristics for Safe Pedestrian Crossings: Developing Warrants for Sustainable Urban Safety"

_sustainability, doi:10.3390/su16104182_

Round 1

Reviewer 1 Report

Comments and Suggestions for Authors

Road accidents involving pedestrians are a serious problem and it is advisable to seek measures to improve pedestrian safety. For this reason, work submitted for publication should be assessed positively. The issues described in the publication concern specific conditions of pedestrian traffic in developing countries. Therefore, the publication will enjoy less interest among European researchers. The conclusions from the described research are not universal. They concern, for example, infrastructure solutions for pedestrians that are considered dangerous in Europe - pedestrian crossings with 3 or 4 lanes without traffic lights.

The authors rightly point out that one of the problems is the inappropriate behavior of drivers and the lack of discipline among pedestrians. However, there is no statement as to whether these behaviors can be improved (corrected) with infrastructure solutions. What is the experience in India so far? This comment should be included in the summary (conclusions).

Some of the cited publications (e.g. 11, 12, 13, 14, 15) concern the period from 30 years ago and require critical commentary - were the authors only concerned with the research methodology or with practical conclusions? This should be commented on.

To better understand the described research, it is advisable to complete information about the legal regulations in force in India regarding pedestrian crossings at designated crossings. It is also appropriate to include in paragraph 3 photos of the test section or its schematic drawing. This can be very helpful when interpreting research results (better understanding them).

In the described case of research, as many as 47% were two- and three-wheeled vehicles, probably moving at low speeds. In such a situation, can the conclusions from the research (proposed criteria for selecting the type of pedestrian crossing) be useful in other cities? This requires a separate comment.

According to the physical interpretation, the relationships in Figs. 5 and 6 should be presented differently, i.e. the dependence of the acceptance time on the speed and the dependence of time losses on the traffic intensity. Figures 5 and 6 could be left as is, but this would require clarification.

A valuable element of the work is the analysis of the criteria for using various forms of pedestrian protection when crossing the road. This advantage determines the recommendation to print the assessed work after minor corrections.

Reviewer 2 Report

Comments and Suggestions for Authors

The objective of this paper is to determine the relationship between vehicle flow and gap acceptance and the relationship between vehicle flow and waiting time by collecting data on vehicle and pedestrian flow at an intersection in India and using the data to study the variables such as vehicle and pedestrian flow, pedestrian acceptance gaps and pedestrian waiting time, through which suitable pedestrian crossing facilities are determined. The logic of the paper is clear and the conclusions are valuable for the design of pedestrian facilities. I believe that the paper can only be accepted if the following questions are revised or answered:

Major concern:

1. The author should give a top view of the data collection intersection as well as a picture of the site.

2. Do all the data come from the same day and is the p-v relationship affected by the day of the week or holiday? Is the data from only one day generalisable?

4. Although the total amount of gap acceptance data was high, it was all from the same intersection and the same time period. Would the gap acceptance still be the same if the intersection was wider or narrower? Is there a difference in gap acceptance between time periods?

5. In the section paragraph of Section 4.2, the authors said “a heavily negatively skewed distribution rather than a normal distribution”, but latter they said “The result of the Lilliefors test indicates that there is no significant difference from the normal distribution”. Which conclusion was right?

Minor issue:

1. In line 70, "Existing Pedestrians Crossing Warrants" under the title should be deleted.

3. ‘ in line 159 should be deleted.

2. Line 255, ‘inn’ should be ‘in’. In addition, Figure 1 needs to clearly refer to what the horizontal and vertical axes represent, respectively. y and x should not be used in the formulas in the figure and should be consistent with the variable names in the context.

3. The two lines in Figure 3 should be of different line types and be accompanied by a legend.

4. Figures 8 and 9 are of poor clarity and the font on the figures should be enlarged.

5. Figures and tables should be centred.
